# Stochastic Proximal Langevin Algorithm: Potential Splitting and Nonasymptotic Rates

**Adil Salim**     **Dmitry Kovalev**     **Peter Richtárik**$^*$
King Abdullah University of Science and Technology, Thuwal, Saudi Arabia

## Abstract

We propose a new algorithm—Stochastic Proximal Langevin Algorithm (SPLA)—for sampling from a log concave distribution. Our method is a generalization of the Langevin algorithm to potentials expressed as the sum of one stochastic smooth term and multiple stochastic nonsmooth terms. In each iteration, our splitting technique only requires access to a stochastic gradient of the smooth term and a stochastic proximal operator for each of the nonsmooth terms. We establish nonasymptotic sublinear and linear convergence rates under convexity and strong convexity of the smooth term, respectively, expressed in terms of the KL divergence and Wasserstein distance. We illustrate the efficiency of our sampling technique through numerical simulations on a Bayesian learning task.

## 1  Introduction

Many applications in the field of Bayesian machine learning require to sample from a probability distribution $\mu^\star$ with density $\mu^\star(x)$, $x \in \mathbb{R}^d$. Due to their scalability, Monte Carlo Markov Chain (MCMC) methods such as Langevin Monte Carlo [48] or Hamiltonian Monte Carlo [28] are popular algorithms to solve such problems. Monte Carlo methods typically generate a sequence of random variables $(x^k)_{k \geq 0}$ with the property that the distribution of $x^k$ approaches $\mu^\star$ as $k$ grows.

While the theory of MCMC algorithms has remained mainly asymptotic, in recent years the exploration of non-asymptotic properties of such algorithms has led to a renaissance in the field [14, 26, 39, 15, 16, 19, 22, 12, 53, 10, 52]. In particular, if $\mu^\star(x) \propto \exp(-U(x))$, where $U$ is a smooth convex function, [14, 19] provide explicit convergence rates for the *Langevin algorithm (LA)*

$$x^{k+1} = x^k - \gamma \nabla U(x^k) + \sqrt{2\gamma} W^k,$$

where $\gamma > 0$ and $(W^k)_{k \geq 0}$ is a sequence of i.i.d. standard Gaussian random variables. The function $U$, also called the *potential*, enters the algorithm through its gradient.

In optimization, the problem $\min U$ where $U$ is composite, *i.e.* is a sum of nonsmooth terms which must be handled separately, has many instances, see [17, Section 2]. These optimization problems can be seen as a Maximum A Posteriori (MAP) computation of some Bayesian model. Sampling a posterori in these models allows for a better Bayesian inference [20]. In these cases, the task of sampling a posterori takes the form of sampling from the target distribution $\mu^\star$, where $U$ has a composite form.

In this work we study the setting where the potential $U$ is the sum of a single smooth and a potentially large number of nonsmooth convex functions. In particular, we consider the problem

$$\text{Sample from} \quad \mu^\star(x) \propto \exp(-U(x)), \quad \text{where} \quad U(x) := F(x) + \sum_{i=1}^{n} G_i(x), \qquad (1)$$

---

$^*$Also affiliated with Moscow Institute of Physics and Technology, Dolgoprudny, Russia.

Table 1: Complexity results obtained in Corollaries 2, 3 and 4 of our main result (Theorem 1).

| $F$ | Stepsize $\gamma$ | Rate | Theorem |
|---|---|---|---|
| convex | $\mathcal{O}(\varepsilon)$ | $\mathrm{KL}(\mu_{\hat{x}^k} \mid \mu^\star) \leq \frac{1}{2\gamma(k+1)} W^2(\mu_{x^0}, \mu^\star) + \mathcal{O}(\gamma)$ | Cor 2 |
| $\alpha$-strongly convex | $\mathcal{O}(\varepsilon\alpha)$ | $W^2(\mu_{x^k}, \mu^\star) \leq (1 - \gamma\alpha)^k W^2(\mu_{x^0}, \mu^\star) + \mathcal{O}\left(\frac{\gamma}{\alpha}\right)$ | Cor 3 |
| $\alpha$-strongly convex | $\mathcal{O}(\varepsilon\alpha)$ | $\mathrm{KL}(\mu_{\tilde{x}^k} \mid \mu^\star) \leq \alpha(1 - \gamma\alpha)^{k+1} W^2(\mu_{x^0}, \mu^\star) + \mathcal{O}(\gamma)$ | Cor 4 |

where $F : \mathbb{R}^d \to \mathbb{R}$ is a smooth convex function and $G_1, \ldots, G_n : \mathbb{R}^d \to \mathbb{R}$ are (possibly nonsmooth) convex functions. The additive model for $U$ offers ample flexibility as typically there are multiple decompositions of $U$ in the form (1).

## 2 Contributions

We now briefly comment some of the key contributions of this work.

$\diamond$ **A splitting technique for Langevin algorithm.** We propose a new variant of LA for solving (1), which we call *Stochastic Proximal Langevin Algorithm (SPLA)*. We assume that $F$ and $G$ can be written as expectations over some simpler functions $f(\cdot, \xi)$ and $g_i(\cdot, \xi)$

$$F(x) = \mathbb{E}_\xi(f(x, \xi)), \quad \text{and} \quad G_i(x) = \mathbb{E}_\xi(g_i(x, \xi)). \tag{2}$$

SPLA (see Algorithm 1 in Section 4) only requires accesses to the gradient of $f(\cdot, \xi)$ and to proximity operators of the functions $g_i(\cdot, \xi)$. SPLA can be seen as a Langevin version of the stochastic Passty algorithm [30, 36]. To the best of our knowledge, this is the first time a splitting technique that involves multiple (stochastic) proximity operators is used in a Langevin algorithm.

*Remarks:* Current forms of LA tackle problem (1) using stochastic subgradients [18]. If $n = 1$ and $G_1$ is proximable (i.e., the learner has access to the full proximity operator of $G_1$), it has recently been proposed to use proximity operators instead of (sub)gradients [20, 18], as it is done in the optimization literature [29, 2]. Indeed, in this case, the proximal stochastic gradient method is an efficient method to minimize $U$. If $n > 1$, and the functions $G_i$ are proximable (but not $U$), the minimization of $U$ is usually tackled using the *operator splitting* framework: the (stochastic) three-operator splitting [51, 17] or (stochastic) primal dual algorithms [13, 46, 9, 35]. These algorithms involve the computation of (stochastic) gradients and (full) proximity operators and enjoy numerical stability properties. However, proximity operators are sometimes difficult to implement. In this case, *stochastic proximity operators* are cheaper[2] than full proximity operators and numerically more stable than stochastic subgradients to handle nonsmooth terms [32, 31, 4, 5, 6] but also smooth [43] terms. In this paper, we bring together the advantages of operator splitting and stochastic proximity operators for sampling purposes.

$\diamond$ **Theory.** We perform a nonasymptotic convergence analysis of SPLA. Our main result, Theorem 1, gives a tractable recursion involving the Kullback-Leibler divergence and Wasserstein distance (when $U$ is strongly convex) between $\mu^\star$ and the distribution of certain samples generated by our method. We use this result to show that the KL divergence is lower than $\varepsilon$ after $\mathcal{O}(1/\varepsilon^2)$ iterations if the constant stepsize $\gamma = \mathcal{O}(\varepsilon)$ is used (Corollary 2). Assuming $F$ is $\alpha$-strongly convex, we show that the Wasserstein distance and (resp. the KL divergence) decrease exponentially, up to an oscillation region of size $\mathcal{O}(\gamma/\alpha)$ (resp. $\mathcal{O}(\gamma)$) as shown in Corollary 3 (resp. Corollary 4). If we wish to push the Wasserstein distance below $\varepsilon$ (resp. the KL divergence below $\alpha\varepsilon$), this could be achieved by setting $\gamma = \mathcal{O}(\varepsilon\alpha)$, and it would be sufficient to take $\mathcal{O}(1/\varepsilon \log 1/\varepsilon)$ iterations. These results are summarized in Table 1. The obtained convergence rates match the previous known results obtained in simpler settings [18]. Note that convergence rates of optimization methods involving multiple stochastic proximity operators haven't been established yet.

*Remarks:* Our proof technique is inspired by [38], which is itself based on [18]. In [38], the authors consider the $n = 1$ case, and assume that the smooth function $F$ is proximable. In [18], a proximal stochastic (sub)gradient Langevin algorithm is studied. In this paper, convergence rates are established by showing that the probability distributions of the iterates shadow some discretized gradient flow defined on a measure space. Hence, our work is a contribution to recent efforts to understand Langevin algorithm as an optimization algorithm in a space of probability measures [26, 49, 3].

⋄ **Online setting.** In online settings, $U$ is unknown but revealed across time. Our approach provides a reasonable algorithm for such situations, especially in cases when the information revealed about $U$ is stationary in time. In particular, this includes online Bayesian learning with structured priors or nonsmooth log likelihood [50, 23, 40, 47]. In this context, the learner is required to sample from some posterior distribution $\mu^\star$ that takes the form (1) where $F, G_1, \ldots, G_n$ are intractable. However, these functions can be cheaply sampled, or are revealed across time through i.i.d. streaming data.

⋄ **Simulations.** We illustrate the promise of our approach numerically by performing experiments with SPLA. We first apply SPLA to a stochastic and nonsmooth toy model with a ground truth. Then, we consider the problem of *sampling from the posterior distribution in the Graph Trend Filtering* context [47]. For this nonsmooth large scale simulation problem, SPLA is performing better than the state of the art method that uses stochastic subgradients instead of stochastic proximity operators. Indeed, in the optimization litterature [2], proximity operators are already known to be more stable than subgradients.

## 3   Technical Preliminaries

In this section, we recall certain notions from convex analysis and probability theory, which are keys to the developments in this paper, state our main assumptions, and introduce needed notations.

### 3.1   Subdifferential, minimal section and proximity operator

Given a convex function $g : \mathbb{R}^d \to \mathbb{R}$, its *subdifferential* at $x$, $\partial g(x)$, is the set

$$\partial g(x) := \left\{ d \in \mathbb{R}^d \ : \ g(x) + \langle d, y - x \rangle \le g(y) \right\}.$$

Since $\partial g(x)$ is a nonempty closed convex set [2], the projection of 0 onto $\partial g(x)$—the least norm element in the set $\partial g(x)$—is well defined, and we call this element $\nabla^0 g(x)$. The function $\nabla^0 g : \mathbb{R}^d \to \mathbb{R}^d$ is called the *minimal section* of $\partial g$. The *proximity operator* associated with $g$ is the mapping $\mathrm{prox}_g : \mathbb{R}^d \to \mathbb{R}^d$ defined by

$$\mathrm{prox}_g(x) := \arg\min_{y \in \mathbb{R}^d} \left\{ \tfrac{1}{2} \|x - y\|^2 + g(y) \right\}.$$

Due to its implicit definition, $\mathrm{prox}_g$ can be hard to evaluate.

### 3.2   Stochastic structure of $F$ and $G_i$: integrability, smoothness and convexity

Here we detail the assumptions behind the stochastic structure (2) of the functions $F = \mathbb{E}_\xi(f(x, \xi))$ and $G_i = \mathbb{E}_\xi(g_i(x, \xi))$ defining the potential $U$. Let $(\Omega, \mathscr{F}, \mathbb{P})$ be a probability space and denote $\mathbb{E}$ the mathematical expectation and $\mathbb{V}$ the variance. Consider $\xi$ a random variable from $\Omega$ to another probability space $(\Xi, \mathscr{G})$ with distribution $\mu$.

**Assumption 1** (Integrability). The functions $f : \mathbb{R}^d \times \Xi \to \mathbb{R}^d$ and $g_i : \mathbb{R}^d \times \Xi \to \mathbb{R}^d$, $i = 1, \ldots, n$, are $\mu$-integrable for every $x \in \mathbb{R}^d$.

Furthermore, we will make the following convexity and smoothness assumptions.

**Assumption 2** (Convexity and differentiability). The function $f(\cdot, s)$ is convex and differentiable for every $s \in \Xi$. The functions $g_i(\cdot, s)$ are convex for every $i \in \{1, 2, \ldots, n\}$.

The gradient of $f(\cdot, s)$ is denoted $\nabla f(\cdot, s)$, the subdifferential of $g_i(\cdot, s)$ is denoted $\partial g_i(\cdot, s)$ and its minimal section is denoted $\nabla^0 g_i(\cdot, s)$. Under Assumption 2, it is known that $F$ is convex and differentiable and that $\nabla F(x) = \mathbb{E}_\xi(\nabla f(x, \xi))$ [34]. Next, we assume that $F$ is smooth and $\alpha$-strongly convex. However, we allow $\alpha = 0$ if $F$ is not strongly convex. We will only assume that $\alpha > 0$ in Corollaries 3 and 4.

**Assumption 3** (Convexity and smoothness of $F$)**.** The gradient of $F$ is $L$-Lipschitz continuous, where $L \geq 0$. Moreover, $F$ is $\alpha$-strongly convex, where $\alpha \geq 0$.

Under Assumption 2, the second part of the above holds for $\alpha = 0$. Finally, we will introduce two noise conditions on the stochastic (sub)gradients of $f(\cdot, s)$ and $g_i(\cdot, s)$.

**Assumption 4** (Bounded variance of $\nabla f(x, \cdot)$)**.** There exists $\sigma_F \geq 0$, such that $\mathbb{V}_\xi(\|\nabla f(x, \xi)\|) \leq \sigma_F^2$ for every $x \in \mathbb{R}^d$.

**Assumption 5** (Bounded second moment of $\nabla^0 g_i(x, \cdot)$)**.** For every $i \in \{1, 2, \ldots, n\}$, there exists $L_{G_i} \geq 0$ such that $\mathbb{E}_\xi(\|\nabla^0 g_i(x, \xi)\|^2) \leq L_{G_i}^2$ for every $x \in \mathbb{R}^d$.

Note that if $g_i(\cdot, s)$ is $\ell_i(s)$-Lipschitz continuous for every $s \in \Xi$, and if $\ell_i(s)$ is $\mu$-square integrable, then Assumption 5 holds.

### 3.3 KL divergence, entropy and potential energy

Recall from (1) that $U := F + \sum_{i=1}^n G_i$ and assume that $\int \exp(-U(x))dx < \infty$. Our goal is to sample from the unique distribution $\mu^\star$ over $\mathbb{R}^d$ with density $\mu^\star(x)$ (w.r.t. the Lebesgue measure denoted $\mathcal{L}$) proportional to $\exp(-U(x))$, for which we write $\mu^\star(x) \propto \exp(-U(x))$. The closeness between the samples of our algorithm and the target distribution $\mu^\star$ will be evaluated in terms of information theoretic and optimal transport theoretic quantities.

Let $\mathcal{B}(\mathbb{R}^d)$ be the Borel $\sigma$-field of $\mathbb{R}^d$. Given two nonnegative measures $\mu$ and $\nu$ on $(\mathbb{R}^d, \mathcal{B}(\mathbb{R}^d))$, we write $\mu \ll \nu$ if $\mu$ is absolutely continuous w.r.t. $\nu$, and denote $\frac{d\mu}{d\nu}$ its density. The *Kullback-Leibler (KL) divergence* between $\mu$ and $\nu$, $\mathrm{KL}(\mu \mid \nu)$, quantifies the closeness between $\mu$ and $\nu$. If $\mu \ll \nu$, then the KL divergence is defined by

$$\mathrm{KL}(\mu \mid \nu) := \int \log \left( \frac{d\mu}{d\nu}(x) \right) d\mu(x),$$

and otherwise we set $\mathrm{KL}(\mu \mid \nu) = +\infty$. Up to an additive constant, $\mathrm{KL}(\cdot \mid \mu^\star)$ can be seen as the sum of two terms [37]: the *entropy* $\mathcal{H}(\mu)$ and the *potential energy* $\mathcal{E}_U(\mu)$. The entropy of $\mu$ is given by $\mathcal{H}(\mu) := \mathrm{KL}(\mu \mid \mathcal{L})$, and the potential energy of $\mu$ is defined by $\mathcal{E}_U(\mu) := \int U d\mu(x)$.

### 3.4 Wasserstein distance

Although the KL divergence is equal to zero if and only if $\mu = \nu$, it is not a mathematical distance (metric). The *Wasserstein distance*, defined below, metrizes the space $\mathcal{P}_2(\mathbb{R}^d)$ of probability measures over $\mathbb{R}^d$ with a finite second moment. Consider $\mu, \nu \in \mathcal{P}_2(\mathbb{R}^d)$. A *transference plan* of $(\mu, \nu)$ is a probability measure $\upsilon$ over $(\mathbb{R}^d \times \mathbb{R}^d, \mathcal{B}(\mathbb{R}^d \times \mathbb{R}^d))$ with marginals $\mu, \nu$ : for every $A \in \mathcal{B}(\mathbb{R}^d)$, $\upsilon(A \times \mathbb{R}^d) = \mu(A)$ and $\upsilon(\mathbb{R}^d \times A) = \nu(A)$. In particular, the product measure $\mu \otimes \nu$ is a transference plan. We denote $\Gamma(\mu, \nu)$ the set of transference plans. A *coupling* of $(\mu, \nu)$ is a random variable $(X, Y)$ over some probability space with values in $(\mathbb{R}^d \times \mathbb{R}^d, \mathcal{B}(\mathbb{R}^d \times \mathbb{R}^d))$ (i.e., $X$ and $Y$ are random variables with values in $\mathbb{R}^d$) such that the distribution of $X$ is $\mu$ and the distribution of $Y$ is $\nu$. In other words, $(X, Y)$ is a coupling of $\mu, \nu$ if the distribution of $(X, Y)$ is a transference plan of $\mu, \nu$. The Wasserstein distance of order 2 between $\mu$ and $\nu$ is defined by

$$W^2(\mu, \nu) := \inf \left\{ \int_{\mathbb{R}^d \times \mathbb{R}^d} \|x - y\|^2 d\upsilon(x, y), \quad \upsilon \in \Gamma(\mu, \nu) \right\}.$$

One can see that $W^2(\mu, \nu) = \inf \mathbb{E}(\|X - Y\|^2)$, where the inf is taken over all couplings $(X, Y)$ of $\mu, \nu$ defined on some probability space with expectation $\mathbb{E}$.

## 4 The SPLA Algorithm and its Convergence Rates

### 4.1 The algorithm

To solve the sampling problem (1), our Stochastic Proximal Langevin Algorithm (SPLA) generates a sequence of random variables $(x^k)_{k \geq 0}$ from $(\Omega, \mathscr{F}, \mathbb{P})$ to $(\mathbb{R}^d, \mathcal{B}(\mathbb{R}^d))$ defined as follows

$$
\begin{aligned}
z^k &= x^k - \gamma \nabla f(x^k, \xi^k) \\
y_0^k &= z^k + \sqrt{2\gamma} W^k \\
y_i^k &= \mathrm{prox}_{\gamma g_i(\cdot, \xi^k)}(y_{i-1}^k) \quad \text{for} \quad i = 1, \ldots, n \\
x^{k+1} &= y_n^k,
\end{aligned}
$$

where $(W^k)_{k \geq 0}$ is a sequence of i.i.d. standard Gaussian random variables, $(\xi^k)_{k \geq 0}$ is a sequence of i.i.d. copies of $\xi$ and $\gamma > 0$ is a positive step size. Our SPLA method is formalized as Algorithm 1; its steps are explained therein.

---

**Algorithm 1** Stochastic Proximal Langevin Algorithm (SPLA)

---

   **Initialize**: $x^0 \in \mathbb{R}^d$
   **for** $k = 0, 1, 2, \ldots$ **do**
      Sample random $\xi^k$          $\triangleright$ used for stoch. approximation: $F \approx f(\cdot, \xi^k)$ and $G_i \approx g_i(\cdot, \xi^k)$
      $z^k = x^k - \gamma \nabla f(x^k, \xi^k)$          $\triangleright$ a stochastic gradient descent step in $F$
      Sample random $W^k$          $\triangleright$ a standard Gaussian vector in $\mathbb{R}^d$
      $y_0^k = z^k + \sqrt{2\gamma} W^k$          $\triangleright$ a Langevin step w.r.t. $F$
      **for** $i = 1, \ldots, n$ **do**
         $y_i^k = \mathrm{prox}_{\gamma g_i(\cdot, \xi^k)}(y_{i-1}^k)$          $\triangleright$ prox step to handle the term $G_i(\cdot) = \mathbb{E}_\xi g_i(\cdot, \xi)$
      **end for**
      $x^{k+1} = y_n^k$          $\triangleright$ the final SPLA step, accounting for $F$ and $G_1, G_2, \ldots, G_n$
   **end for**

---

### 4.2 Main theorem

We now state our main results in terms of Kullback-Leibler divergence and Wasserstein distance. We denote $\mu_x$ the distribution of every random variable $x$ defined on $(\Omega, \mathscr{F}, \mathbb{P})$.

**Theorem 1.** Let Assumptions 1–5 hold and assume that $\gamma \leq 1/L$. There exists $C \geq 0$ such that,

$$
2\gamma \, \mathrm{KL}(\mu_{y_0^k} \mid \mu^\star) \leq (1 - \gamma\alpha) W^2(\mu_{x^k}, \mu^\star) - W^2(\mu_{x^{k+1}}, \mu^\star) + \gamma^2 (2\sigma_F^2 + 2Ld + C). \quad (3)
$$

The constant $C$ can be expressed as a linear combination of $L_{G_1}^2, \ldots, L_{G_n}^2$ with integer coefficients. Moreover, if $n = 2$, then $C := 2(L_{G_1}^2 + L_{G_2}^2)$. More generally, if for every $i \in \{2, \ldots, n\}$, $g_i(\cdot, \xi)$ admits almost surely the representation $g_i(\cdot, \xi) = \tilde{g}_i(\cdot, \xi_i)$ where $\xi_2, \ldots, \xi_n$ are independent random variables, then $C := n \sum_{i=1}^{n} L_{G_i}^2$.

*Proof.* A full proof can be found in the Supplementary material. We only sketch the main steps here. For every $\mu$-integrable function $g : \mathbb{R}^d \to \mathbb{R}$, we denote $\mathcal{E}_g(\mu) = \int g \mathrm{d}\mu$. Moreover, we denote $\mathcal{F} = \mathcal{E}_U + \mathcal{H}$. First, using [18, Lemma 1], $\mu^\star \in \mathcal{P}_2(\mathbb{R}^d)$, $\mathcal{E}_U(\mu^\star), \mathcal{H}(\mu^\star) < \infty$ and if $\mu \in \mathcal{P}_2(\mathbb{R}^d)$, then

$$
\mathrm{KL}(\mu \mid \mu^\star) = \mathcal{E}_U(\mu) + \mathcal{H}(\mu) - (\mathcal{E}_U(\mu^\star) + \mathcal{H}(\mu^\star)) = \mathcal{F}(\mu) - \mathcal{F}(\mu^\star),
$$

provided that $\mathcal{E}_U(\mu) < \infty$. Then, we decompose $\mathcal{E}_U(\mu) = \mathcal{E}_F(\mu) + \mathcal{E}_G(\mu)$ where $G = \sum_{i=1}^{n} G_i$. Using [18] again, we can establish the inequality

$$
2\gamma \left[ \mathcal{H}(\mu_{y_0^k}) - \mathcal{H}(\mu^\star) \right] \leq W^2(\mu_{z^k}, \mu^\star) - W^2(\mu_{y_0^k}, \mu^\star). \quad (4)
$$

Then, if $\gamma \leq 1/L$ we obtain, for every random variable $a$ with distribution $\mu^\star$,

$$
\mathbb{E}\left[ \left\| z^k - a \right\|^2 \right] \leq (1 - \gamma\alpha) \mathbb{E}\left[ \left\| x^k - a \right\|^2 \right] + 2\gamma \left[ \mathcal{E}_F(\mu^\star) - \mathcal{E}_F(\mu_{z^k}) \right] + 2\gamma^2 \sigma_F^2, \quad (5)
$$

using standard computations regarding the stochastic gradient descent algorithm. Using the smoothness of $F$ and the definition of the Wasserstein distance, this implies

$$2\gamma\left[\mathcal{E}_F(\mu_{y_0^k}) - \mathcal{E}_F(\mu^\star)\right] \leq (1 - \gamma\alpha)W^2(\mu_{x^k}, \mu^\star) - W^2(\mu_{z^k}, \mu^\star) + \gamma^2(2\sigma_F^2 + 2Ld).$$

It remains to establish $2\gamma\left[\mathcal{E}_G(\mu_{y_0^k}) - \mathcal{E}_G(\mu^\star)\right] \leq W^2(\mu_{y_0^k}, \mu^\star) - W^2(\mu_{x^{k+1}}, \mu^\star) + \gamma^2C$, which is the main technical challenge of the proof. This is done using the frameworks of Yosida approximation of random subdifferentials and Moreau regularizations of random convex functions [2]. Equation (3) is obtained by summing the obtained inequalities. □

## 4.3 Link with Wasserstein Gradient Flows

Equation (3) is reminiscent of the fact that the SPLA shadows the gradient flow of $\mathrm{KL}(\cdot \mid \mu^\star)$ in the metric space $(\mathcal{P}_2(\mathbb{R}^d), W)$. To see this, first consider the gradient flow associated to $F$. By definition, it is the flow of the differential equation [7]

$$\frac{\mathrm{d}}{\mathrm{d}t}\mathsf{x}(t) = -\nabla F(\mathsf{x}(t)), \quad t > 0. \tag{6}$$

The function $\mathsf{x}$ can alternatively be defined as a solution of the variational inequalities

$$2\left(F(\mathsf{x}(t)) - F(a)\right) \leq -\frac{\mathrm{d}}{\mathrm{d}t}\|\mathsf{x}(t) - a\|^2, \quad t > 0, \quad \forall a \in \mathbb{R}^d. \tag{7}$$

The iterates $(u^k)_{k \geq 0}$ of the stochastic gradient descent (SGD) algorithm applied to $F$ can be seen as a (noisy) Euler discretization of (6) with a step size $\gamma > 0$. This idea has been used successfully in the stochastic approximation litterature [33, 24]. This analogy goes further since a fundamental inequality used to analyze SGD applied to $F$ is ([27])

$$2\gamma\mathbb{E}\left(F(u^{k+1}) - F(a)\right) \leq \mathbb{E}\|u^k - a\|^2 - \mathbb{E}\|u^{k+1} - a\|^2 + \gamma^2 K, \quad k \geq 0,$$

where $K \geq 0$ is some constant, which can be seen as a discrete counterpart of (7). Note that this inequality is similar to (5) that is used in the proof of Theorem 1.

In the optimal transport theory, the point of view of (7) is used to define the gradient flow of a (geodesically) convex function $\mathcal{F}$ defined on $\mathcal{P}_2(\mathbb{R}^d)$ (see [37] or [1, Page 280]). Indeed, the gradient flow $(\nu_t)_{t \geq 0}$ of $\mathcal{F}$ in the space $(\mathcal{P}_2(\mathbb{R}^d), W)$ satisfies for every $t > 0, \mu \in \mathcal{P}_2(\mathbb{R}^d)$,

$$2\left(\mathcal{F}(\nu_t) - \mathcal{F}(\mu)\right) \leq -\frac{\mathrm{d}}{\mathrm{d}t}W^2(\nu_t, \mu), \tag{8}$$

which can be seen as a continuous time counterpart of Equation (3) by setting $\mathcal{F} = \mathrm{KL}(\cdot \mid \mu^\star)$. Furthermore, Equation (4) in the proof of Theorem 1 is also related to (8). It is obtained by applying Equation (8) with $\mathcal{F} = \mathcal{H}$ and $\nu_0 = \mu_{z^k}$ (see *e.g* [18, Lemma 5]).

## 4.4 Explicit convergence rates for convex and strongly convex $F$

Corollaries 2, 3 and 4 below are obtained by unrolling the recursion provided by Theorem 1. The results are summarized in Table 1.

**Corollary 2** (Convex $F$). Consider a sequence of independent random variables $(j_k)_{k \geq 0}$ such that $(j_k)_{k \geq 0}$ is independent of $(W^k)_k$ and $(\xi^k)_k$, and the distribution of $j_k$ is uniform over $\{0, \ldots, k\}$. Denote $\hat{x}^k = y_0^{j_k}$. If $\gamma \leq 1/L$, then,

$$\mathrm{KL}(\mu_{\hat{x}^k} \mid \mu^\star) \leq \frac{1}{2\gamma(k+1)}W^2(\mu_{x^0}, \mu^\star) + \frac{\gamma}{2}(2\sigma_F^2 + 2Ld + C).$$

Hence, given any $\varepsilon > 0$, choosing stepsize $\gamma = \min\left\{\frac{1}{L}, \frac{\varepsilon}{2\sigma_F^2 + 2Ld + C}\right\}$ and a number of iterations

$$k + 1 \geq \max\left\{\frac{L}{\varepsilon}, \frac{2\sigma_F^2 + 2Ld + C}{\varepsilon^2}\right\} W^2(\mu_{x^0}, \mu^\star),$$

implies $\mathrm{KL}(\mu_{\hat{x}^k} \mid \mu^\star) \leq \varepsilon$.

**Corollary 3** (Strongly convex $F$). If $\alpha > 0$ and $\gamma \leq 1/L$, then,

$$W^2(\mu_{x^k}, \mu^\star) \leq (1 - \gamma\alpha)^k W^2(\mu_{x^0}, \mu^\star) + \frac{\gamma(2\sigma_F^2 + 2Ld + C)}{\alpha}.$$

Hence, given any $\varepsilon > 0$, choosing stepsize $\gamma = \min\left\{\frac{1}{L}, \frac{\varepsilon\alpha}{2(2\sigma_F^2 + 2Ld + C)}\right\}$ and a number of iterations

$$k \geq \max\left\{\frac{L}{\alpha}, \frac{2(2\sigma_F^2 + 2Ld + C)}{\varepsilon\alpha^2}\right\} \log\left(\frac{2W^2(\mu_{x^0}, \mu^\star)}{\varepsilon}\right),$$

implies $W^2(\mu_{x^k}, \mu^\star) \leq \varepsilon$.

**Corollary 4** (Strongly convex $F$)**.** Consider a sequence of independent random variables $(j_k)_{k \geq 0}$ such that $(j_k)_k$ is independent of $(W^k)_k$ and $(\xi^k)_k$. Assume that the distribution of $j_k$ is geometric over $\{0, \ldots, k\}$:

$$\mathbb{P}(j_k = r) \propto (1 - \gamma\alpha)^{-r}.$$

Denote $\tilde{x}^k = x^{j_k}$. If $\alpha > 0$ and $\gamma \leq 1/L$, then,

$$\mathrm{KL}(\mu_{\tilde{x}^k} \mid \mu^\star) \leq \frac{\alpha W^2(\mu_{x^0}, \mu^\star)}{2} \cdot \frac{(1 - \gamma\alpha)^{k+1}}{1 - (1 - \gamma\alpha)^{k+1}} + \frac{\gamma(2\sigma_F^2 + 2Ld + C)}{2}.$$

Hence, given any $\varepsilon > 0$, choosing stepsize $\gamma = \min\left\{\frac{1}{L}, \frac{\varepsilon\alpha}{2\sigma_F^2 + 2Ld + C}\right\}$ and a number of iterations

$$k \geq \max\left\{\frac{L}{\alpha}, \frac{2\sigma_F^2 + 2Ld + C}{\varepsilon\alpha^2}\right\} \log\left(2\max\left\{1, \frac{W^2(\mu_{x^0}, \mu^\star)}{\varepsilon}\right\}\right),$$

implies $\mathrm{KL}(\mu_{\tilde{x}^k} \mid \mu^\star) \leq \alpha\varepsilon$.

We can compare these bounds with the one of [18]. First, in the particular case $n = 1$ and $g_1(\cdot, s) \equiv G_1$, SPLA boils down to the algorithm of [18, Section 4.2], Corollary 2 matches exactly [18, Corollary 18] and Corollary 3 matches [18, Corollary 22]. To our knowledge, Corollary 4 has no counterpart in the litterature. We now focus on the case $F \equiv 0$ and $n = 1$ of SPLA, as it concentrates the innovations of our paper. In this case, $L = 0$ and $\sigma_F = 0$. Compared to the Stochastic Subgradient Langevin Algorithm (SSLA) [18, Section 4.1], Corollary 2 matches with [18, Corollary 14].

## 5 Numerical experiments

### 5.1 Simulations using a ground truth

We first concentrate on the case $F \equiv 0$ and $n = 1$. Let $U = |x| = \mathbb{E}_\xi(|x| + x\xi)$ $(g_1(x, s) = |x| + xs)$, where $\xi$ is standard Gaussian. The target $\mu^\star \propto \exp(-U)$ is a standard Laplace distribution in $\mathbb{R}$. In this case, $L = \alpha = \sigma_F = 0$ and $C = L_{G_1}^2 = 2$. We shall illustrate the bound on $\mathrm{KL}(\mu_{\hat{x}^k} \mid \mu^\star)$ (Corollary 2 for SPLA and [18, Corollary 14] for SSLA) for both algorithms using histograms. Note that the distribution $\mu_{\hat{x}^k}$ of $\hat{x}^k$ is a (deterministic) mixture of the $\mu_{x^j}$: $\mu_{\hat{x}^k} = \frac{1}{k}\sum_{j=1}^k \mu_{x^j}$. Using Pinsker inequality, we can bound the total variation distance between $\mu_{\hat{x}^k}$ and $\mu^\star$ from the bound on KL, and illustrate this by histograms. In Figure 1, we take $\gamma = 10$ and do $10^5$ iterations of both algorithms. Note that here the complexity of one iteration of SPLA or SSLA is the same. One can

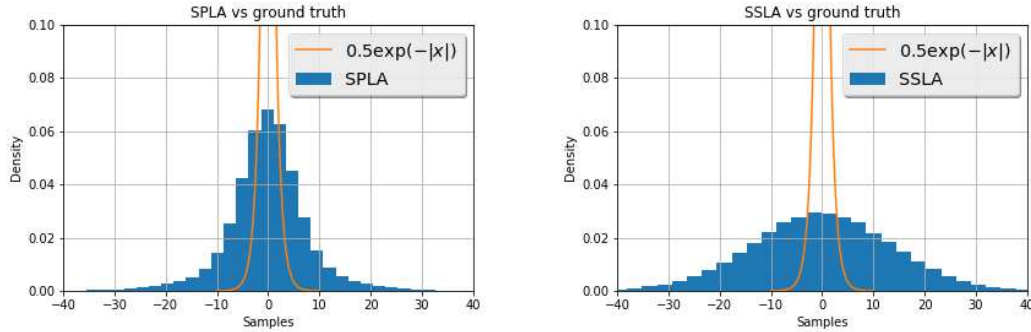

Figure 1: Comparison between histograms of SPLA and SSLA and the true density $0.5\exp(-|x|)$.

see that SPLA enjoys the well known advantages of stochastic proximal methods [43]: precision, numerical stability (less outliers), and robustness to step size.

---

**Algorithm 2** SPLA for the Graph Trend Filtering

---

**Initialize**: $x^0 \in \mathbb{R}^V$
**for** $k = 0, 1, 2, \ldots$ **do**
    $z^k = x^k - \frac{\gamma}{\sigma^2}(x^k - Y)$
    Sample random $W^k$                              ▷ standard Gaussian vector in $\mathbb{R}^V$
    $y_0^k = z^k + \sqrt{2\gamma}W^k$
    **for** $i = 1, \ldots, n$ **do**
        Sample uniform random edges $e_i$
        $y_i^k = \text{prox}_{\gamma g_{e_i}}(y_{i-1}^k)$
    **end for**
    $x^{k+1} = y_n^k$
**end for**

---

## 5.2   Application to Trend Filtering on Graphs

In this section we consider the following Bayesian point of view of *trend filtering on graphs* [42]. Consider a finite undirected graph $G = (V, E)$, where $V$ is the set of vertices and $E$ is the set of edges. Denote $d$ the cardinality of $V$ and $|E|$ the cardinality of $E$. A realization of a random vector $Y \in \mathbb{R}^V$ is observed. In a Bayesian framework, the distribution of $Y$ is parametrized by a vector $X \in \mathbb{R}^V$ which is itself random and whose distribution $p$ is proportional to $\exp(-\lambda \, \text{TV}(x, G))$, where $\lambda > 0$ is a scaling parameter and where for every $x \in \mathbb{R}^V$

$$\text{TV}(x, G) = \sum_{i,j \in V, \{i,j\} \in E} |x(i) - x(j)|,$$

is the Total Variation regularization over $G$. The goal is to learn $X$ after an observation of $Y$. The paper [47] consider the case where the distribution of $Y$ given $X$ (a.k.a the likelihood) is proportional to $\exp(-\frac{1}{2\sigma^2}\|X - y\|^2)$, where $\sigma \geq 0$ is another scaling parameter. In other words, the distribution of $Y$ given $X$ is $N(X, \sigma^2 I)$, a normal distribution centered at $X$ with variance $\sigma^2 I$ (where $I$ is the $d \times d$ identity matrix). Denoting

$$\pi(x \mid y) \propto \exp(-U(x)), \quad U(x) = \tfrac{1}{2\sigma^2}\|x - y\|^2 + \lambda \, \text{TV}(x, G),$$

the posterior distribution of $X$ given $Y$, the maximum a posteriori estimator in this Bayesian framework is called the Graph Trend Filtering estimate [47]. It can be written

$$x^\star = \arg\max_{x \in \mathbb{R}^V} \pi(x \mid Y) = \arg\min_{x \in \mathbb{R}^V} \tfrac{1}{2\sigma^2}\|x - Y\|^2 + \lambda \, \text{TV}(x, G).$$

Although maximum a posteriori estimators carry some information, they are not able to capture uncertainty in the learned parameters. Samples a posteriori provide a better understanding of the posterior distribution and allow to compute other Bayesian estimates such as confidence intervals. This allows to avoid overfitting among other things. In our context, sampling a posteriori would require to sample from the target distribution $\mu^\star(x) = \pi(x \mid Y)$.

In the case where $G$ is a 2D grid (which can be identified with an image), the proximity operator of $\text{TV}(\cdot, G)$ can be computed using a subroutine [8] and the proximal stochastic gradient Langevin algorithm can be used to sample from $\pi(\cdot \mid Y)$ [20, 18]. However, on a large/general graph, the proximity operator of $\text{TV}(\cdot, G)$ is hard to evaluate [41, 36]. Since $\text{TV}(\cdot, G)$ is written as a sum, we shall rather select a batch of random edges and compute the proximity operators over these randomly chosen edges. More precisely, we write the potential $U$ defining $\pi(x \mid Y)$ in the form (1) by setting

$$U(x) = F(x) + \sum_{i=1}^{n} G_i(x), \quad F(x) = \tfrac{1}{2\sigma^2}\|x - Y\|^2, \quad G_i(x) = \lambda\tfrac{|E|}{n}\mathbb{E}_{e_i}\left(|x(v_i) - x(w_i)|\right),$$

where for every $i \in \{1, \ldots, n\}$, $e_i = \{v_i, w_i\} \in E$ is an uniform random edge and the $e_i$ are independent. For every edge $e = \{v, w\} \in E$, (where $v, w$ are vertices) denote $g_e(x) = \lambda\frac{|E|}{n}|x(v) - x(w)|$ and note that $G_i(x) = \mathbb{E}_{e_i}(g_{e_i}(x))$. The parameter $n$ can be seen as a batch parameter: $\sum_{i=1}^{n} g_{e_i}(x)$ is an unbiaised approximation of $TV(x, G)$. Also note that we set $f(\cdot, s) \equiv F$. The SPLA applied to sample from $\pi(\cdot \mid Y)$ is presented as Algorithm 2. In our simulations, the SPLA is compared to two different versions of the Langevin algorithm. In the

Stochastic Subgradient Langevin Algorithm (SSLA) [18], stochastic subgradients of $g_{e_i}$ are used instead of stochastic proximity operators. In the Proximal Langevin Algorithm (ProxLA) [18], the full proximity operator of $\sum_{i=1}^{n} G_i$ is computed using a subroutine. As mentioned in [36, 47], we use the gradient algorithm for the dual problem. The plots in Figure 2 provide simulations of the algorithms on our machine (using one thread of a 2,800 MHz CPU and 256GB RAM). Additional numerical experiments are available in the Appendix. Four real life graphs from the dataset [25] are considered : the Facebook graph (4,039 nodes and 88,234 edges, extracted from the Facebook social network), the Youtube graph (1,134,890 nodes and 2,987,624 edges, extracted from the social network included in the Youtube website), the Amazon graph (the 334,863 nodes represent products linked by and 925,872 edges) and the DBLP graph (a co-authorship network of 317,080 nodes and 1,049,866 edges). On the larger graphs, we only simulate SPLA and SSLA since the computation of a full proximity operator becomes prohibitive. Numerical experiments over the Amazon and the DBLP graphs are available in the Supplementary material.

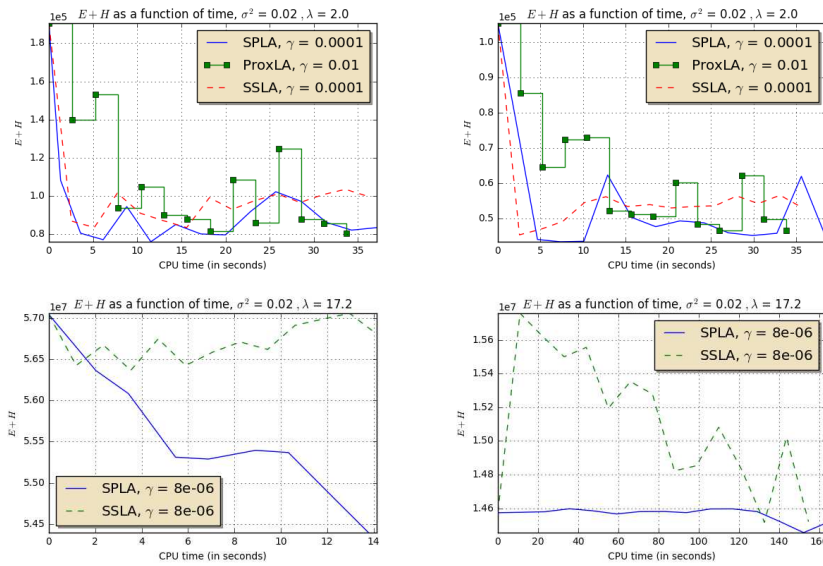

Figure 2: **Top row:** The functional $\mathcal{F} = \mathcal{H} + \mathcal{E}_U$ as a function of CPU time for the three algorithms over the Facebook graph. Left: $Y \sim N(0, I)$. Right: $Y \sim N(0, I)$ and then half of the coordinates of $Y$ are put to zero. **Bottom row:** The functional $\mathcal{F} = \mathcal{H} + \mathcal{E}_U$ as a function of CPU time for the two algorithms over the Youtube graph. Left: $Y \sim N(0, I)$. Right: $Y \sim N(0, I)$ and then half of the coordinates of $Y$ are put to zero.

In our simulations, we represent the functional $\mathcal{F} = \mathcal{H} + \mathcal{E}_U$ as a function of CPU time while running the algorithms. The parameters $\lambda$ and $\sigma$ are chosen such that the log likelihood term and the Total Variation regularization term have the same weight. The functionals $\mathcal{H}$ and $\mathcal{E}_U$ are estimated using five random realizations of each iterate $\hat{x}^k$ ($\mathcal{H}$ is estimated using a kernel density estimator). The batch parameter $n$ is equal to $400$. We consider cases where $Y$ has a standard gaussian distribution and cases where half of the components of $Y$ are standard gaussians and half are equal to zero (this correspond to the graph inpainting task [11]). SPLA and SSLA are always simulated with the same step size.

As expected, the numerical experiments show the advantage of using stochastic proximity operators instead of stochastic subgradients. It is a standard fact that proximity operators are better than subgradients to tackle $\ell^1$-norm terms [2]. Our figures show that stochastic proximity operators are numerically more stable than alternatives [43]. Our figures also show the advantage of stochastic methods (SSLA or SPLA) over deterministic ones for large scale problems. The SSLA and the SPLA provide iterates about one hundred times more frequently than ProxLA, and are faster in the first iterations.

## Footnotes

[2]See www.proximity-operator.net

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
