[Supplementary Material · langevin19.pdf]

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

&emsp;Sample random $W^k$ &emsp;&emsp;&emsp;&emsp;&emsp;&emsp;&emsp;&emsp;&emsp;&emsp; ▷ standard Gaussian vector in $\mathbb{R}^V$
&emsp;$y_0^k = z^k + \sqrt{2\gamma}W^k$
&emsp;**for** $i = 1, \ldots, n$ **do**
&emsp;&emsp;Sample uniform random edges $e_i$
&emsp;&emsp;$y_i^k = \text{prox}_{\gamma g_{e_i}}(y_{i-1}^k)$
&emsp;**end for**
&emsp;$x^{k+1} = y_n^k$
**end for**

---

## 5.2 Application to Trend Filtering on Graphs

In this section we consider the following Bayesian point of view of *trend filtering on graphs* [42]. Consider a finite undirected graph $G = (V, E)$, where $V$ is the set of vertices and $E$ is the set of edges. Denote $d$ the cardinality of $V$ and $|E|$ the cardinality of $E$. A realization of a random vector $Y \in \mathbb{R}^V$ is observed. In a Bayesian framework, the distribution of $Y$ is parametrized by a vector $X \in \mathbb{R}^V$ which is itself random and whose distribution $p$ is proportional to $\exp(-\lambda \, \text{TV}(x, G))$, where $\lambda > 0$ is a scaling parameter and where for every $x \in \mathbb{R}^V$

$$\text{TV}(x, G) = \sum_{i,j \in V, \{i,j\} \in E} |x(i) - x(j)|,$$

is the Total Variation regularization over $G$. The goal is to learn $X$ after an observation of $Y$. The paper [47] consider the case where the distribution of $Y$ given $X$ (a.k.a the likelihood) is proportional to $\exp(-\frac{1}{2\sigma^2}\|X - y\|^2)$, where $\sigma \geq 0$ is another scaling parameter. In other words, the distribution of $Y$ given $X$ is $N(X, \sigma^2 I)$, a normal distribution centered at $X$ with variance $\sigma^2 I$ (where $I$ is the $d \times d$ identity matrix). Denoting

$$\pi(x \mid y) \propto \exp(-U(x)), \quad U(x) = \tfrac{1}{2\sigma^2}\|x - y\|^2 + \lambda \, \text{TV}(x, G),$$

the posterior distribution of $X$ given $Y$, the maximum a posteriori estimator in this Bayesian framework is called the Graph Trend Filtering estimate [47]. It can be written

$$x^\star = \underset{x \in \mathbb{R}^V}{\arg\max} \, \pi(x \mid Y) = \underset{x \in \mathbb{R}^V}{\arg\min} \, \

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

# Appendix

## Contents

## A    Lemmas Needed for the Proof of the Main Theorem

In order to prove Theorem 1, we will need to establish several lemmas. For every convex function $g : \mathbb{R}^d \to \mathbb{R}$, we denote $\mathcal{E}_g(\mu) = \int g \mathrm{d}\mu$. In the sequel, we assume that Assumptions 1–5 hold true.

We first recall [18, Lemma 1].

**Lemma 5.** The target distribution satisfies $\mu^\star \in \mathcal{P}_2(\mathbb{R}^d), \mathcal{E}_U(\mu^\star)$ and $\mathcal{H}(\mu^\star) < \infty$.

Moreover, if $\mu \in \mathcal{P}_2(\mathbb{R}^d)$, then

$$\mathrm{KL}(\mu \mid \mu^\star) = \mathcal{E}_U(\mu) + \mathcal{H}(\mu) - (\mathcal{E}_U(\mu^\star) + \mathcal{H}(\mu^\star)) = \mathcal{F}(\mu) - \mathcal{F}(\mu^\star),$$

provided that $\mathcal{E}_U(\mu) < \infty$.

**Lemma 6.**

$$2\gamma \left[ \mathcal{H}(\mu_{y_0^k}) - \mathcal{H}(\mu^\star) \right] \leq W^2(\mu_{z^k}, \mu^\star) - W^2(\mu_{y_0^k}, \mu^\star). \tag{9}$$

*Proof.* This is an application of [18, Lemma 5] with $\mu \leftarrow \mu_{z^k}$ and $\pi \leftarrow \mu^\star$.

The proof is the roughly the same as proving the $\mathcal{O}(1/k)$ convergence rate of the gradient descent algorithm, but in continuous time and in the space $\mathcal{P}_2(\mathbb{R}^d)$. For the sake of completeness, we provide the main arguments of a different proof than [18] here, that makes easier the connection with Lyapunov techniques used in the analysis of gradient descent/gradient flows in Euclidean spaces.

Consider Brownian motion $(B_t)$ and the rescaled Brownian Motion $(\sqrt{2}B_t)$ initialized with $\sqrt{2}B_0 \sim \mu_{z^k}$ and denote, for every $t \geq 0$, $\nu_t$ the distribution of $\sqrt{2}B_t$. Then $(\nu_t)_t$ is a gradient flow of $\mathcal{H}$ (see [37]). This implies (see [1, Page 280])

$$\forall t > 0, \quad 2\left(\mathcal{H}(\nu_t) - \mathcal{H}(\mu^\star)\right) \leq -\frac{\mathrm{d}}{\mathrm{d}t} W^2(\nu_t, \mu^\star). \tag{10}$$

This also implies [45, Page 711] that (the objective function) $\mathcal{H}$ is a Lyapunov function for (the gradient flow) $(\nu_t)_{t \geq 0}$ :

$$\forall t > 0, \quad \frac{\mathrm{d}}{\mathrm{d}t} \mathcal{H}(\nu_t) \leq 0. \tag{11}$$

Now consider the function $\ell(t) = t\left(\mathcal{H}(\nu_t) - \mathcal{H}(\mu^\star)\right) + \frac{1}{2} W^2(\nu_t, \mu^\star)$. For every $t > 0$,

$$\frac{\mathrm{d}}{\mathrm{d}t}\ell(t) = \left(\mathcal{H}(\nu_t) - \mathcal{H}(\mu^\star)\right) + t\frac{\mathrm{d}}{\mathrm{d}t}\mathcal{H}(\nu_t) - \left(\mathcal{H}(\nu_t) - \mathcal{H}(\mu^\star)\right) \leq 0,$$

using the inequalities (A) and (A). In other words, $\ell$ is also a Lyapunov function. Therefore, for every $\varepsilon > 0, \ell(\gamma) \leq \ell(\varepsilon)$, which implies[3] $\ell(\gamma) \leq \ell(0)$ *i.e*,

$$\gamma \left(\mathcal{H}(\nu_\gamma) - \mathcal{H}(\mu^\star)\right) \leq \frac{1}{2} W^2(\nu_0, \mu^\star) - \frac{1}{2} W^2(\nu_\gamma, \mu^\star).$$

It remains to note that $\nu_0 = \mu_{z^k}$ and $\nu_\gamma = \mu_{y_0^k}$.    □

**Lemma 7.**

$$2\gamma \left[ \mathcal{E}_F(\mu_{y_0^k}) - \mathcal{E}_F(\mu_{z^k}) \right] \leq 2\gamma^2 Ld . \tag{12}$$

*Proof.* This is an application of [18, Lemma 3] with $\mu \leftarrow \mu_{z^k}$. We provide the proof for the sake of completeness. It can be related to smoothing techniques used in optimization [21].

First, using the convexity and the smoothness of $F$,

$$0 \leq F(y_0^k) - F(z^k) + \left\langle \nabla F(z^k), z^k - y_0^k \right\rangle \leq \frac{L}{2}\|y_0^k - z^k\|^2. \tag{13}$$

Note that $y_0^k - z^k = \sqrt{2\gamma}W^k$ is independent of $z^k$, that $\mathbb{E}(y_0^k - z^k) = 0$ and that $\mathbb{E}(\|y_0^k - z^k\|^2) = 2\gamma d$. Taking the expectation in (A) gives the result.    □

**Lemma 8.** Let $\gamma \leq 1/L$. Then

$$2\gamma \left[ \mathcal{E}_F(\mu_{z^k}) - \mathcal{E}_F(\mu^\star) \right] \leq (1 - \gamma\alpha)W^2(\mu_{x^k}, \mu^\star) - W^2(\mu_{z^k}, \mu^\star) + 2\gamma^2\sigma_F^2. \tag{14}$$

*Proof.* We choose arbitrary $a \in \mathbb{R}^d$ and start with an upper-bound for $\left\| z^k - a \right\|^2$:

$$\left\| z^k - a \right\|^2 = \left\| x^k - a - \gamma\nabla f(x^k, \xi^k) \right\|^2$$
$$= \left\| x^k - a \right\|^2 + 2\gamma \left\langle \nabla f(x^k, \xi^k), a - x_k \right\rangle + \gamma^2 \left\| \nabla f(x^k, \xi^k) \right\|^2.$$

By taking an expectation with respect to $\xi^k$ we get

$$\mathbb{E}_{\xi^k} \left[ \left\| z^k - a \right\|^2 \right] = \left\| x^k - a \right\|^2 + 2\gamma \left\langle \nabla F(x^k), a - x^k \right\rangle + \gamma^2 \left\| \nabla F(x^k) \right\|^2$$
$$+ \gamma^2 \mathbb{E}_{\xi^k} \left[ \left\| \nabla F(x^k) - \nabla f(x^k, \xi^k) \right\|^2 \right]$$
$$\leq \left\| x^k - a \right\|^2 + 2\gamma \left( F(a) - F(x^k) - \frac{\alpha}{2} \left\| x^k - a \right\|^2 \right) + \gamma^2 \left\| \nabla F(x^k) \right\|^2 + \gamma^2\sigma_F^2$$

$$= (1 - \gamma\alpha) \left\| x^k - a \right\|^2 + 2\gamma \left( F(a) - F(x^k) \right) + \gamma^2 \left\| \nabla F(x^k) \right\|^2 + \gamma^2\sigma_F^2. \tag{15}$$

Next, we use the $L$-smoothness of $F$ and the fact that $\gamma \leq 1/L$:

$$\mathbb{E}_{\xi^k} F(z^k) \leq F(x^k) + \mathbb{E}_{\xi^k} \left[ \left\langle \nabla F(x^k), z^k - x^k \right\rangle \right] + \frac{L}{2} \mathbb{E}_{\xi^k} \left[ \left\| z^k - x^k \right\|^2 \right]$$
$$= F(x^k) - \gamma \left( 1 - \frac{L\gamma}{2} \right) \left\| \nabla F(x^k) \right\|^2 + \frac{L}{2}\gamma^2 \mathbb{E}_{\xi^k} \left[ \left\| \nabla f(x^k, \xi^k) \right\|^2 - \left\| \nabla F(x^k) \right\|^2 \right]$$
$$\leq F(x^k) - \frac{\gamma}{2} \left\| \nabla F(x^k) \right\|^2 + \frac{L}{2}\gamma^2\sigma_F^2.$$

Since $L\gamma \leq 1$,

$$\mathbb{E}_{\xi^k} F(z^k) \leq F(x^k) - \frac{\gamma}{2} \left\| \nabla F(x^k) \right\|^2 + \frac{\gamma}{2}\sigma_F^2,$$

which gives us an upper-bound for $\gamma^2 \left\| \nabla F(x^k) \right\|^2$:

$$\gamma^2 \left\| \nabla F(x^k) \right\|^2 \leq 2\gamma \left( F(x^k) - \mathbb{E}_{\xi^k} F(z^k) \right) + \gamma^2\sigma_F^2,$$

since $\gamma \leq 1/L$. Plugging this into (A) we obtain

$$\mathbb{E}_{\xi^k} \left[ \left\| z^k - a \right\|^2 \right] \leq (1 - \gamma\alpha) \left\| x^k - a \right\|^2 + 2\gamma \left( F(a) - \mathbb{E}_{\xi^k} F(z^k) \right) + 2\gamma^2\sigma_F^2. \tag{16}$$

Now, let $a \sim \mu^\star$, i.e. $a$ is a random vector sampled from the distribution with density $\mu^\star$. By taking the full expectation in (A) we get

$$\mathbb{E} \left[ \left\| z^k - a \right\|^2 \right] \leq (1 - \gamma\alpha)\mathbb{E} \left[ \left\| x^k - a \right\|^2 \right] + 2\gamma \left[ \mathcal{E}_F(\mu^\star) - \mathcal{E}_F(\mu_{z^k}) \right] + 2\gamma^2\sigma_F^2.$$

Using the definition of Wasserstein distance we get

$$W^2(\mu_{z^k}, \mu^\star) \leq (1 - \gamma\alpha)\mathbb{E} \left[ \left\| x^k - a \right\|^2 \right] + 2\gamma \left[ \mathcal{E}_F(\mu^\star) - \mathcal{E}_F(\mu_{z^k}) \right] + 2\gamma^2\sigma_F^2$$

Note that in the last inequality, $x^k$ can be replaced by any random variable with distribution $\mu_{x^k}$. Taking the $\inf$ over all possible couplings of $\mu_{x^k}$ and $\mu^\star$ we get

$$W^2(\mu_{z^k}, \mu^\star) \leq (1 - \gamma\alpha)W^2(\mu_{x^k}, \mu^\star) + 2\gamma \left[ \mathcal{E}_F(\mu^\star) - \mathcal{E}_F(\mu_{z^k}) \right] + 2\gamma^2\sigma_F^2.$$

$\square$

**Remark 1.** We now recall standard facts from convex analysis that will be used without mention in the sequel. These results can be found in [2] or [7]. Let $g : \mathbb{R}^d \to \mathbb{R}$ be a convex function. The Moreau envelope $g^\gamma$ of $g$ is defined by

$$g^\gamma(x) = \min_{y \in \mathbb{R}^d} g(y) + \frac{1}{2\gamma} \left\| y - x \right\|^2,$$

and is a $1/\gamma$-smooth convex function such that $g^\gamma(x) \le g(x)$ and $g^\gamma(x) \to_{\gamma \to 0} g(x)$ for every $x \in \mathbb{R}^d$. The proximity operator of $g$ and the Moreau envelope are linked through their definitions

$$g^\gamma(x) = \frac{1}{2\gamma} \left\| \text{prox}_{\gamma g}(x) - x \right\|^2 + g(\text{prox}_{\gamma g}(x)),$$

but also through the relation

$$\text{prox}_{\gamma g}(x) = x - \gamma \nabla g^\gamma(x).$$

The function $\nabla g^\gamma$ is called the Yosida approximation of $\partial g$. The proximity operator $\text{prox}_{\gamma g}$ is 1-Lipschitz continuous, and so is $\gamma \nabla g^\gamma$. The Yosida approximation satisfies moreover

$$\nabla g^\gamma(x) \in \partial g \left( \text{prox}_{\gamma g}(x) \right),$$

for every $x \in \mathbb{R}^d$. Since $g$ only takes finite values, for every $x \in \mathbb{R}^d$, $\partial g(x) \ne \emptyset$. Furthermore, the Yosida approximation satisfies for every $x \in \mathbb{R}^d$,

$$\left\| \nabla g^\gamma(x) \right\| \le \left\| \nabla^0 g(x) \right\|.$$

**Lemma 9.** Let $g : \mathbb{R}^d \to \mathbb{R}$ be a convex function. Then,

$$g^\gamma(x) \ge g(x) - \frac{\gamma}{2} \left\| \nabla^0 g(x) \right\|^2.$$

*Proof.* Let $x \in \mathbb{R}^d$. Using the convexity of $g$ we have for every $y \in \mathbb{R}^d$,

$$g(y) + \frac{1}{2\gamma} \|y - x\|^2 \ge g(x) + \left\langle \nabla^0 g(x), y - x \right\rangle + \frac{1}{2\gamma} \|y - x\|^2. \tag{17}$$

We conclude the proof by taking the minimum over $y$ on both sides of (A). $\square$

**Lemma 10.** For every $i \in \{1, \dots, n\}$, let $g_i : \mathbb{R}^d \to \mathbb{R}$ a convex function. Consider $a, y_0, y_1, \dots, y_n \in \mathbb{R}^d$ such that for every $k = 1, \dots, n$, $y_k = \text{prox}_{\gamma g_k}(y_{k-1})$. Then,

$$\|y_n - a\|^2 \le \|y_0 - a\|^2 - 2\gamma \sum_{k=1}^{n} (g_k^\gamma(y_{k-1}) - g_k(a)).$$

*Proof.* Iterating Equation 1, we have for every $i \in \{1, \dots, n\}$:

$$y_i = y_0 - \gamma \sum_{j=1}^{i} \nabla g_j^\gamma(y_{j-1}).$$

Therefore,

$$\|y_n - a\|^2 = \|y_0 - a\|^2 - 2\gamma \left\langle \sum_{i=1}^{n} \nabla g_i^\gamma(y_{i-1}), y_0 - a \right\rangle + \gamma^2 \left\| \sum_{i=1}^{n} \nabla g_i^\gamma(y_{i-1}) \right\|^2. \tag{18}$$

Since $\nabla g_i^\gamma(y_{i-1}) \in \partial g_i(y_i)$,

$$\langle \nabla g_i^\gamma(y_{i-1}), y_0 - a \rangle = \langle \nabla g_i^\gamma(y_{i-1}), y_i - a \rangle + \gamma \|\nabla g_i^\gamma(y_{i-1})\|^2 + \gamma \left\langle \nabla g_i^\gamma(y_{i-1}), \sum_{j=1}^{i-1} \nabla g_j^\gamma(y_{j-1}) \right\rangle$$

$$\ge g_i(y_i) - g_i(a) + \gamma \|\nabla g_i^\gamma(y_{i-1})\|^2 + \gamma \left\langle \nabla g_i^\gamma(y_{i-1}), \sum_{j=1}^{i-1} \nabla g_j^\gamma(y_{j-1}) \right\rangle.$$

Furthermore,

$$-2\gamma\left\langle\sum_{i=1}^{n}\nabla g_i^{\gamma}(y_{i-1}),y_0-a\right\rangle+\gamma^2\left\|\sum_{i=1}^{n}\nabla g_i^{\gamma}(y_{i-1})\right\|^2$$

$$=-2\gamma\sum_{i=1}^{n}(g_i(y_i)-g_i(a))-2\gamma^2\sum_{i=1}^{n}\left\|\nabla g_i^{\gamma}(y_{i-1})\right\|^2$$

$$+\gamma^2\left\|\sum_{i=1}^{n}\nabla g_i^{\gamma}(y_{i-1})\right\|^2-2\gamma^2\sum_{i=1}^{n}\left\langle\nabla g_i^{\gamma}(y_{i-1}),\sum_{j=1}^{i-1}\nabla g_j^{\gamma}(y_{j-1})\right\rangle.$$

Expanding the square norm $\gamma^2\left\|\sum_{i=1}^{n}\nabla g_i^{\gamma}(y_{i-1})\right\|^2$, all the cross products vanish with the last term. It remains only

$$-2\gamma\left\langle\sum_{i=1}^{n}\nabla g_i^{\gamma}(y_{i-1}),y_0-a\right\rangle+\gamma^2\left\|\sum_{i=1}^{n}\nabla g_i^{\gamma}(y_{i-1})\right\|^2$$

$$=-2\gamma\sum_{i=1}^{n}(g_i(y_i)-g_i(a))-\gamma^2\sum_{i=1}^{n}\left\|\nabla g_i^{\gamma}(y_{i-1})\right\|^2.$$

Since $g_i^{\gamma}(y_{i-1})=\frac{1}{2\gamma}\left\|y_i-y_{i-1}\right\|^2+g_i(y_i)$,

$$-2\gamma\left\langle\sum_{i=1}^{n}\nabla g_i^{\gamma}(y_{i-1}),y_0-a\right\rangle+\gamma^2\left\|\sum_{i=1}^{n}\nabla g_i^{\gamma}(y_{i-1})\right\|^2=-2\gamma\sum_{i=1}^{n}(g_i^{\gamma}(y_{i-1})-g_i(a)).$$

Plugging the last equation into (A) gives the result. $\qquad\square$

**Lemma 11.** There exists $C\geq 0$ which can be expressed as a linear combination of $L_{G_1}^2,\ldots,L_{G_n}^2$ with integer coefficients such that

$$2\gamma\sum_{i=1}^{n}\left[\mathcal{E}_{G_i}(\mu_{y_0^k})-\mathcal{E}_{G_i}(\mu^{\star})\right]\leq W^2(\mu_{y_0^k},\mu^{\star})-W^2(\mu_{x^{k+1}},\mu^{\star})+\gamma^2 C. \tag{19}$$

Moreover, if, for every $i\in\{2,\ldots,n\}$, $g_i(\cdot,\xi)$ admits almost surely the representation $g_i(\cdot,\xi)=\tilde{g}_i(\cdot,\xi_i)$ where $\xi_2,\ldots,\xi_n$ are independent random variables, then one can set $C:=n\sum_{i=1}^{n}L_{G_i}^2$.

*Proof.* Using the convexity of $g_i^{\gamma}(\cdot,\xi^k)$ and Lemma 9,

$$g_i^{\gamma}(y_{i-1}^k,\xi^k)\geq g_i^{\gamma}(y_0^k,\xi^k)+\left\langle\nabla g_i^{\gamma}(y_0^k,\xi^k),y_{i-1}^k-y_0^k\right\rangle$$

$$\geq g_i(y_0^k,\xi^k)-\frac{\gamma}{2}\left\|\nabla^0 g_i(y_0^k,\xi^k)\right\|^2+\left\langle\nabla g_i^{\gamma}(y_0^k,\xi^k),y_{i-1}^k-y_0^k\right\rangle$$

We now look at the last term at the right hand side. If $i=1$ it is equal to zero. If $i\geq 2$, using Young's inequality,

$$-2\gamma\left\langle\nabla g_i^{\gamma}(y_0^k,\xi^k),y_{i-1}^k-y_0^k\right\rangle=\sum_{j=1}^{i-1}-2\left\langle\gamma\nabla g_i^{\gamma}(y_0^k,\xi^k),y_j^k-y_{j-1}^k\right\rangle$$

$$\leq(i-1)\left\|\gamma\nabla g_i^{\gamma}(y_0^k,\xi^k)\right\|^2+\sum_{j=1}^{i-1}\left\|y_j^k-y_{j-1}^k\right\|^2.$$

Combining the two last inequalities with Lemma 10 applied to $y_i\leftarrow y_i^k$ and $g_i\leftarrow g_i(\cdot,\xi^k)$, we have

$$\left\|y_n^k-a\right\|^2\leq\left\|y_0^k-a\right\|^2 \tag{20}$$

$$-2\gamma\sum_{i=1}^{n}(g_i(y_0^k,\xi^k)-g_i(a,\xi^k))$$

$$+\gamma^2\sum_{i=1}^{n}\left\|\nabla^0 g_i(y_0^k,\xi^k)\right\|^2+\sum_{i=2}^{n}\left((i-1)\left\|\gamma\nabla g_i^{\gamma}(y_0^k,\xi^k)\right\|^2+\sum_{j=1}^{i-1}\left\|y_j^k-y_{j-1}^k\right\|^2\right).$$

We now consider two cases. First, assume that for every $i \in \{2, \ldots, n\}$, $g_i(\cdot, \xi)$ admits almost surely the representation $g_i(\cdot, \xi) = \tilde{g}_i(\cdot, \xi_i)$ where $\xi_2, \ldots, \xi_n$ are independent random variables. In this case, $\xi_j^k$ – the $k^{th}$ i.i.d copy of $\xi_j$ – is independent of $y_{j-1}^k$ for every $j \in \{1, \ldots, n\}$. Noting that $y_j^k - y_{j-1}^k = -\gamma \nabla \tilde{g}_j{}^\gamma(y_{j-1}^k, \xi_j^k)$ and using Assumption 5

$$\mathbb{E}\left(\left\|\nabla \tilde{g}_j{}^\gamma(y_{j-1}^k, \xi_j^k)\right\|^2 \mid y_{j-1}^k\right) = \mathbb{E}\left(\left\|\nabla \tilde{g}_j{}^\gamma(\cdot, \xi_j^k)\right\|^2\right)(y_{j-1}^k) = \mathbb{E}\left(\left\|\nabla g_j^\gamma(\cdot, \xi^k)\right\|^2\right)(y_{j-1}^k) \le L_{G_j}^2.$$

Taking a full expectation,

$$\gamma^2 \mathbb{E}\left(\sum_{i=1}^n \left\|\nabla^0 \tilde{g}_i(y_0^k, \xi_i^k)\right\|^2 + \left((i-1)\left\|\gamma \nabla \tilde{g}_i{}^\gamma(y_0^k, \xi_i^k)\right\|^2 + \sum_{j=1}^{i-1}\left\|y_j^k - y_{j-1}^k\right\|^2\right)\right) \le \gamma^2 n \sum_{i=1}^n L_{G_i}^2. \tag{21}$$

In this case we set $C := n \sum_{i=1}^n L_{G_i}^2$. Obviously, this cover the case $n = 2$.

Second (general case), denote $\beta_j = \mathbb{E}\left\|y_j^k - y_{j-1}^k\right\|^2$. We write

$$\nabla g_j^\gamma(y_{j-1}^k, \xi^k) = \nabla g_j^\gamma(y_0^k, \xi^k) + \left(\nabla g_j^\gamma(y_{j-1}^k, \xi^k) - \nabla g_j^\gamma(y_{j-2}^k, \xi^k)\right) + \ldots + \left(\nabla g_j^\gamma(y_1^k, \xi^k) - \nabla g_j^\gamma(y_0^k, \xi^k)\right).$$

Using Young's inequality and the fact that $\gamma \nabla g_j^\gamma(\cdot, \xi^k)$ is 1-Lipschitz continous,

$$\beta_j \le j\left(L_{G_j}^2 + \beta_{j-1} + \ldots + \beta_1\right).$$

Noting that $\beta_1 \le L_{G_1}^2$, it is easy to prove (by induction) that there exists a linear combination of the $L_{G_1}^2, \ldots, L_{G_n}^2$ with integer coefficients denoted $C \ge 0$ such that

$$\gamma^2 \mathbb{E}\left(\sum_{i=1}^n \left\|\nabla^0 g_i(y_0^k, \xi^k)\right\|^2 + \left((i-1)\left\|\gamma \nabla g_i^\gamma(y_0^k, \xi^k)\right\|^2 + \sum_{j=1}^{i-1}\left\|y_j^k - y_{j-1}^k\right\|^2\right)\right) \le \gamma^2 C. \tag{22}$$

Finally, taking the expectation in (A) and plugging (A) or (A),

$$\mathbb{E}\left\|y_n^k - a\right\|^2 \le \mathbb{E}\left\|y_0^k - a\right\|^2$$
$$- 2\gamma \sum_{i=1}^n \left(\mathbb{E}(G_i(y_0^k)) - \mathbb{E}(G_i(a))\right) + \gamma^2 C.$$

Using the definition of $\mathcal{E}_{G_i}$ and taking the inf over all couplings $y_0^k, a$ of $\mu_{y_0^k}, \mu^\star$, we get

$$W^2(\mu_{x^{k+1}}, \mu^\star) = W^2(\mu_{y_n^k}, \mu^\star) \le W^2(\mu_{y_0^k}, \mu^\star) + 2\gamma \sum_{i=1}^n \left[\mathcal{E}_{G_i}(\mu^\star) - \mathcal{E}_{G_i}(\mu_{y_0^k})\right] + \gamma^2 C.$$

$\square$

# B Proof of Theorem 1

By summing up (8) and (7) we get

$$2\gamma \left[ \mathcal{E}_F(\mu_{y_0^k}) - \mathcal{E}_F(\mu^\star) \right] \leq (1 - \gamma\alpha) W^2(\mu_{x^k}, \mu^\star) - W^2(\mu_{z^k}, \mu^\star) + \gamma^2(2\sigma_F^2 + 2Ld).$$

Adding (6) leads to

$$2\gamma \left[ \mathcal{E}_F(\mu_{y_0^k}) - \mathcal{E}_F(\mu^\star) + \mathcal{H}(\mu_{y_0^k}) - \mathcal{H}(\mu^\star) \right] \leq (1 - \gamma\alpha) W^2(\mu_{x^k}, \mu^\star) - W^2(\mu_{y_0^k}, \mu^\star)$$
$$+ \gamma^2(2\sigma_F^2 + 2Ld).$$

Finally, by adding (11) we get

$$2\gamma \left[ \mathcal{E}_F(\mu_{y_0^k}) - \mathcal{E}_F(\mu^\star) + \mathcal{H}(\mu_{y_0^k}) - \mathcal{H}(\mu^\star) + \sum_{i=1}^{n} \left( \mathcal{E}_{G_i}(\mu_{y_0^k}) - \mathcal{E}_{G_i}(\mu^\star) \right) \right]$$
$$\leq (1 - \gamma\alpha) W^2(\mu_{x^k}, \mu^\star) - W^2(\mu_{x^{k+1}}, \mu^\star) + \gamma^2(2\sigma_F^2 + 2Ld + C),$$

which, along with Lemma 5, concludes the proof.

## C Proof of Corollary 2

From (3), for all $j = 0, \ldots, k$ we get

$$2\gamma \left[ \mathcal{F}(\mu_{y_0^j}) - \mathcal{F}(\mu^\star) \right] \le W^2(\mu_{x^j}, \mu^\star) - W^2(\mu_{x^{j+1}}, \mu^\star) + \gamma^2 (2\sigma_F^2 + 2Ld + C). \tag{23}$$

Summing up (C) for $j = 0, \ldots, k$ leads to

$$2\gamma \sum_{j=0}^{k} \left[ \mathcal{F}(\mu_{y_0^j}) - \mathcal{F}(\mu^\star) \right] \le W^2(\mu_{x^0}, \mu^\star) - W^2(\mu_{x^{k+1}}, \mu^\star) + \gamma^2 (k+1)(2\sigma_F^2 + 2Ld + C)$$

$$\le W^2(\mu_{x^0}, \mu^\star) + \gamma^2 (k+1)(2\sigma_F^2 + 2Ld + C).$$

Using Lemma 5 and the convexity of KL divergence [44, Theorem 11], $\mathcal{F}$ is convex on $\mathcal{P}_2(\mathbb{R}^d)$. Since $\mu_{\hat{x}^k} = \frac{1}{k+1} \sum_{j=0}^{k} \mu_{y_0^j}$,

$$\mathcal{F}(\mu_{\hat{x}^k}) \le \frac{1}{k+1} \sum_{j=0}^{k} \mathcal{F}(\mu_{y_0^j}),$$

hence

$$\mathcal{F}(\mu_{\hat{x}^k}) - \mathcal{F}(\mu^\star) \le \frac{1}{2\gamma(k+1)} W^2(\mu_{x^0}, \mu^\star) + \frac{\gamma}{2}(2\sigma_F^2 + 2Ld + C).$$

Hence, given any $\varepsilon > 0$, choosing stepsize $\gamma = \min \left\{ \frac{1}{L}, \frac{\varepsilon}{2\sigma_F^2 + 2Ld + C} \right\}$ leads to

$$\frac{\gamma}{2}(2\sigma_F^2 + 2Ld + C) \le \frac{\varepsilon}{2}.$$

If the number of iterations is

$$k + 1 \ge \max \left\{ \frac{L}{\varepsilon}, \frac{2\sigma_F^2 + 2Ld + C}{\varepsilon^2} \right\} W^2(\mu_{x^0}, \mu^\star),$$

then,

$$\frac{1}{2\gamma(k+1)} W^2(\mu_{x^0}, \mu^\star) \le \frac{\varepsilon}{2}.$$

This implies $\mathcal{F}(\mu_{\hat{x}^k}) - \mathcal{F}(\mu^\star) \le \varepsilon$, and the proof is concluded by applying Lemma 5.

## D   Proof of Corollary 3

From (5), $\mathcal{F}(\mu_{y_0^j}) \geq \mathcal{F}(\mu^\star)$. From (3), for all $j = 0, \ldots, k-1$ we get

$$W^2(\mu_{x^{j+1}}, \mu^\star) \leq (1 - \gamma\alpha)W^2(\mu_{x^j}, \mu^\star) + \gamma^2(2\sigma_F^2 + 2Ld + C).$$

After unrolling this recurrence we get

$$W^2(\mu_{x^k}, \mu^\star) \leq (1 - \gamma\alpha)^k W^2(\mu_{x^0}, \mu^\star) + \gamma^2(2\sigma_F^2 + 2Ld + C)\sum_{j=0}^{k-1}(1 - \gamma\alpha)^j$$

$$= (1 - \gamma\alpha)^k W^2(\mu_{x^0}, \mu^\star) + \gamma^2(2\sigma_F^2 + 2Ld + C)\frac{1 - (1 - \gamma\alpha)^k}{\gamma\alpha}$$

$$\leq (1 - \gamma\alpha)^k W^2(\mu_{x^0}, \mu^\star) + \frac{\gamma(2\sigma_F^2 + 2Ld + C)}{\alpha}.$$

The first part is proven. Setting $\gamma = \min\left\{\frac{1}{L}, \frac{\varepsilon\alpha}{2(2\sigma_F^2 + 2Ld + C)}\right\}$ gives

$$W^2(\mu_{x^k}, \mu^\star) \leq (1 - \gamma\alpha)^k W^2(\mu_{x^0}, \mu^\star) + \frac{\varepsilon}{2}.$$

If

$$k \geq \frac{1}{\gamma\alpha}\log\left(\frac{2W^2(\mu_{x^0}, \mu^\star)}{\varepsilon}\right),$$

then,

$$(1 - \gamma\alpha)^k W^2(\mu_{x^0}, \mu^\star) \leq \frac{\varepsilon}{2},$$

which concludes the proof.

# E  Proof of Corollary 4

From (3), for all $j = 0, \ldots, k$ we get

$$2\gamma \left[ \mathcal{F}(\mu_{y_0^j}) - \mathcal{F}(\mu^\star) \right] \leq (1 - \gamma\alpha) W^2(\mu_{x^j}, \mu^\star) - W^2(\mu_{x^{j+1}}, \mu^\star) + \gamma^2 (2\sigma_F^2 + 2Ld + C). \quad (24)$$

By dividing (E) by $(1 - \gamma\alpha)^j$ we get

$$\frac{2\gamma}{(1-\gamma\alpha)^j} \left[ \mathcal{F}(\mu_{y_0^j}) - \mathcal{F}(\mu^\star) \right] \leq \frac{W^2(\mu_{x^j}, \mu^\star)}{(1-\gamma\alpha)^{j-1}} - \frac{W^2(\mu_{x^{j+1}}, \mu^\star)}{(1-\gamma\alpha)^j} + \frac{\gamma^2(2\sigma_F^2 + 2Ld + C)}{(1-\gamma\alpha)^j}. \quad (25)$$

Summing up (E) for $j = 0, \ldots, k$ gives

$$\sum_{j=0}^{k} \frac{2\gamma}{(1-\gamma\alpha)^j} \left[ \mathcal{F}(\mu_{y_0^j}) - \mathcal{F}(\mu^\star) \right] \leq (1-\gamma\alpha) W^2(\mu_{x^0}, \mu^\star) - \frac{W^2(\mu_{x^{k+1}}, \mu^\star)}{(1-\gamma\alpha)^k} + \sum_{j=0}^{k} \frac{\gamma^2(2\sigma_F^2 + 2Ld + C)}{(1-\gamma\alpha)^j}$$

$$\leq (1-\alpha\gamma) W^2(\mu_{x^0}, \mu^\star) + \sum_{j=0}^{k} \frac{\gamma^2(2\sigma_F^2 + 2Ld + C)}{(1-\gamma\alpha)^j}.$$

Using Lemma 5 and the convexity of KL divergence [44, Theorem 11], $\mathcal{F}$ is convex on $\mathcal{P}_2(\mathbb{R}^d)$. Since

$$\mu_{\tilde{x}^k} = \sum_{j=0}^{k} \frac{(1-\gamma\alpha)^{-j}}{\sum_{r=0}^{k}(1-\gamma\alpha)^{-r}} \mu_{x^j},$$

$$\mathcal{F}(\mu_{\tilde{x}^k}) \sum_{j=0}^{k}(1-\gamma\alpha)^{-j} = \sum_{j=0}^{k}(1-\gamma\alpha)^{-j} \mathcal{F}(\mu_{\tilde{x}^k}) \leq \sum_{j=0}^{k}(1-\gamma\alpha)^{-j} \mathcal{F}(\mu_{x^j}),$$

hence

$$2\gamma \sum_{j=0}^{k}(1-\gamma\alpha)^{-j} \left[ \mathcal{F}(\mu_{\tilde{x}^k}) - \mathcal{F}(\mu^\star) \right] \leq (1-\alpha\gamma) W^2(\mu_{x^0}, \mu^\star) + \sum_{j=0}^{k} \frac{\gamma^2(2\sigma_F^2 + 2Ld + C)}{(1-\gamma\alpha)^j}.$$

After dividing by $2\gamma \sum_{j=0}^{k}(1-\gamma\alpha)^{-j}$ we obtain

$$\mathcal{F}(\mu_{\tilde{x}^k}) - \mathcal{F}(\mu^\star) \leq \frac{W^2(\mu_{x^0}, \mu^\star)}{2\gamma \sum_{j=0}^{k}(1-\gamma\alpha)^{-(j+1)}} + \frac{\gamma(2\sigma_F^2 + 2Ld + C)}{2}. \quad (26)$$

Now, we perform a simplification of the sum:

$$\sum_{j=0}^{k} \gamma(1-\gamma\alpha)^{-(j+1)} = \frac{\gamma}{(1-\gamma\alpha)} \sum_{j=0}^{k}(1-\gamma\alpha)^{-j} = \frac{\gamma}{(1-\gamma\alpha)} \cdot \frac{(1-\gamma\alpha)^{-(k+1)} - 1}{(1-\gamma\alpha)^{-1} - 1}$$

$$= \frac{(1-\gamma\alpha)^{-(k+1)} - 1}{\alpha}.$$

Plugging this into (E) gives

$$\mathcal{F}(\mu_{\tilde{x}^k}) - \mathcal{F}(\mu^\star) \leq \frac{\alpha W^2(\mu_{x^0}, \mu^\star)}{2((1-\gamma\alpha)^{-(k+1)} - 1)} + \frac{\gamma(2\sigma_F^2 + 2Ld + C)}{2}$$

$$= \alpha \left[ \frac{W^2(\mu_{x^0}, \mu^\star)}{2} \cdot \frac{(1-\gamma\alpha)^{k+1}}{1 - (1-\gamma\alpha)^{k+1}} + \frac{\gamma(2\sigma_F^2 + 2Ld + C)}{2\alpha} \right].$$

If $\gamma = \min \left\{ \frac{1}{L}, \frac{\varepsilon\alpha}{2\sigma_F^2 + 2Ld + C} \right\}$ and

$$k \geq \max \left\{ \frac{L}{\alpha}, \frac{2\sigma_F^2 + 2Ld + C}{\varepsilon\alpha^2} \right\} \log \left( 2 \max \left\{ 1, \frac{W^2(\mu_{x^0}, \mu^\star)}{\varepsilon} \right\} \right),$$

then $k \geq \frac{1}{\gamma\alpha} \log 2$. Moreover, $(1 - \gamma\alpha)^{k+1} \leq 1/2$,

$$\mathcal{F}(\mu_{\tilde{x}^k}) - \mathcal{F}(\mu^\star) \leq \alpha \left[ (1 - \gamma\alpha)^{k+1} W^2(\mu_{x^0}, \mu^\star) + \frac{\gamma(2\sigma_F^2 + 2Ld + C)}{2\alpha} \right],$$

and

$$\mathcal{F}(\mu_{\tilde{x}^k}) - \mathcal{F}(\mu^\star) \leq \alpha \left[ (1 - \gamma\alpha)^{k+1} W^2(\mu_{x^0}, \mu^\star) + \frac{\varepsilon}{2} \right].$$

The conclusion follows from Lemma 5.

# F  Additional Numerical Experiments

Figure 3: **Top row:** The functional $\mathcal{F} = \mathcal{H} + \mathcal{E}_U$ as a function of CPU time for the Amazon graph with $Y \sim N(0, I)$. Left: SSLA and SPLA. Right: Only SPLA. **Bottom row:** The functional $\mathcal{F} = \mathcal{H} + \mathcal{E}_U$ as a function of CPU time for the DBLP graph with $Y \sim N(0, I)$. Left: SSLA and SPLA. Right : Only SPLA.

These numerical experiments are conducted over the Amazon and the DBLP graphs. The left curves show the numerical stability of the proximal method (SPLA) with respect to the subgradient method (SSLA). The right curves are zoomed in view of the behavior of SPLA during the same experiments. It can be seen that SPLA still decreases the KL divergence.