[Reviews · NeurIPS 2019]

Reviewer 1



Update: I have read the reports from the other referees and the response from authors. The authors have done a good job addressing the overall concerns, and with the addition of comparisons to the baseline truth, I have changed my assessment to an 8. Originality: It is clear that the method is inspired from the the Passty algorithm, including the proofs therein. The originality is in using a Langevin dynamics over the usual Passty algorithm. Quality: The results are clear and well explained, and the example demonstrates the theoretical results as well. Significance: Overall, the article seems useful, particularly due to the possibility of sequential implementation.

Reviewer 2



** General comments The paper provides an original theoretical contribution which should have impacts at least in methodological developments. It is not clear to me to what extent the necessity to have access to a proximal operator (routinely) represents a serious bottleneck. How of a limiting factor is that? ** Major comments/questions include: - the paper presentation could be improved, see remarks below. Some English mistakes too. - however, the contribution is clearly mentioned, even though a more explicit presentation of the main results of [17] would be helpful. At a technical level - how can bounds involving a random number of iterations be useful in practice? This is also a problem in Table 1 where those rv are not defined. A remark on fact that \mu^\ast is compared to a random measure would be helpful. Is that standard in the related bounds found in the literature (eg [17])? - why are the KL bounds for convex F and strongly convex F involving two different stage of a SPLA transition? For convex F the KL is between \mu^\ast and the r.v. before the proximal step while for strongly convex its between \mu^\ast and an entire SPLA transition. An insight would be very helpful. - check below some proof-related questions. - for the application part: were the Author able to find an application of thm 1 and other corollaries by identifying the constants L, \alpha and C? Also, instead/or in addition of numerical results it would be more useful to compare directly the bounds of SPLA with bounds of other subgradient LA algorithm (if applicable). ** Minor points Check the text around Eq. 1: the first part should be in the text do not use where in series. l26 the proposed additive model for U offers and check the end of that sentence. l54 rather than certain samples perhaps the proba. measures of a specific collection/or subsequence of random variables generated by our method. l67-68 it would be useful to have this comment in a separate remark environment, to highlight this point. l78 if more than one method, name it otherwise correct to methods that use'' l128 entropy is not well defined l142 title, perhaps convergence rates instead of rate l166 comments of this paragraph and especially the last part of it should be moved to a dedicated section as they bring the subtle part of the work. l173 applies to l378 =\int g d\mu not )\int g d\mu l382 the statement is not trivial since in 17 Lemma 4 the inequality is for the function \mathcal{E}_{U} and not \mathcal{H} check ref. [36] Lemma 6, to reach \mu_{y_0^k}, isn't it required to take t=2\gamma instead of t=\gamma? Indeed, \nu_\gamma has the same law as z^k+\sqrt(\gamma) W^k which is not precisely the law of y_0^k. But this would change the inequality, with a factor 4\gamma instead of 2\gamma in [9]. Can you please check that? l443 the factor \gamma^2 should be inside the sum if the sum also applies to the 2nd and last terms ; otherwise a sum is missing for the 2nd and last terms. l449 Why is Lemma 5 required here? l477 it is not clear how the measure \cal{F}(\mu_{y_0^j}) is substituted by \cal{F}(\mu_{\tilde{x}^k}) Application l197 it could be useful to identify the functions f and g_i and the rv \xi of the expectations of functions F and G in this example. Esp. f\equiv f(x) and in this case there is no randomness. l214 v and w are not defined, are they vertices?

Reviewer 3



Response to the rebuttal: Thanks to the authors for responding to my questions. I still feel the experiments of the paper can benefit (especially from the rebuttal) from more work but keeping the theoretical contributions I have decided to increase the score to 7. ========== The authors propose a new sampling algorithm, called Stochastic Proximal Langevin Algorithm, and analyze it for sampling from log-concave distributions where the log density is a sum of smooth and differentiable function along with a sum of non-smooth convex functions. They establish the rates of convergence for the algorithm, linear when the smooth part is strongly convex and sub-linear otherwise (the rates are not surprising). The authors provide a numerical comparison of their algorithm with a few other algorithms for trend filtering on large graphs, demonstrate that for this set-up their algorithm is useful when the full proximal operator is computationally prohibitive and in general it performs faster than Stochastic Subgradient Langevin Algorithm. On technical aspects, their analysis appears to be borrowing heavily from [17] (as the authors have correctly acknowledged). I would like to remark that the paper is well-written and easy to read. I have a few remarks: — Given the introduction of the new algorithm, I believe the work can be motivated better if the authors provide several concrete set-ups/examples where their algorithm can be potentially useful (with some rough gains that one can obtain). Even for the example, they considered, the significance of sampling in comparison to optimization (simply finding the MAP) is not clearly spelled out (other than noting that it might be useful). — It is not clear to me when is SPLA faster than SSLA? and vice-versa? — There should be some trade-off of splitting the non-smooth part into an arbitrary sum. In simple words given f+g, I should not gain by treating g as g/2 + g/2. — I would recommend the authors to provide some clear explanations for computational trade-offs when such splitting is beneficial, and some intuition as to? Moreover, do we gain/lose if we use different noise draws for each proximal split? Is the gain simply analogous to the gains of SGD over GD when the number of samples is large (or is there something more to it)? A paragraph providing intuitions about the new algorithm would make the paper more insightful.

[Author Response · NeurIPS 2019]

We thank to Reviewers 1, 2 and 3 (who gave us marks 7, 8 and 6, respectively) for their pertinent remarks.

**R1+R3: Contribution.** We agree that the optimization counterpart of SPLA can be related to Passty algorithm. However, it is a much more general optimization algorithm than Passty since all the functions are allowed to be expectations treated through stochastic gradients/stochastic prox. Therefore, we call it **Stochastic Passty**. In the optimization literature, the non asymptotic theory of this algorithm is still unknown. The only known particular cases are $n = 1$ and $G_1$ deterministic (prox-SGD), and the case $F = 0$ and $n = 1$ (Stochastic Proximal Point Algorithm, [29]). We were inspired by the proof structure of [17] (we will update this reference as it is published in JMLR), which is very adaptive. It allows to separate the analysis of SPLA into two pieces: the analysis of the optimization counterpart (here, Stochastic Passty) and the analysis of the Gaussian noise. Here, all the non asymptotic analysis of Stochastic Passty had to be done and involves modern tools of convex analysis such as random prox and random subdifferentials.

**R1+R2+R3: Corollaries.** We agree that we could provide more insights on the corollaries (Cor). As suggested by R2 and R3, we can compare the bounds with the one of [17]. First, in the particular case $n = 1$ and $G_1$ deterministic, SPLA boils down to the algorithm of [17, Section 4.2], Cor2 matches exactly Cor18 of [17] and Cor3 matches Cor22[1]. Cor4 has no counterpart in [17]. We now focus on the case $F = 0$ and $n = 1$ of SPLA, as it concentrates the innovations of our work. In this case, $L = 0$ and $\sigma_F = 0$. Compared to SSLA, our Cor2 matches with Cor14 of [17]. Actually, our constant $C$ in Cor2 might be better because $C = L_{G_1}^2 \leq M^2 + D^2$ [1], due to the fact that we only need to bound the L2 norm of the minimal section (and not of any subgradient as in [17])[1]. In summary, [17] only covers the case $n = 1$ and $G_1$ deterministic of Cor2 and Cor3, and doesn't cover Cor4. The main advantage of SPLA over SSLA is its numerical stability (because SPLA is a proximal method [41], see the next paragraph).

**R1+R2: Simulation.** We agree that we could improve the experimental section by using a ground truth. We will add the following comparison of SSLA and SPLA in the case $F = 0$ and $n = 1$. Let $U = |x| = \mathbb{E}_\xi(|x| + x\xi)$ ($g_1(x, s) = |x| + xs$), where $\xi$ is standard Gaussian. $\mu^\star \propto \exp(-U)$ is a standard Laplace distribution in $\mathbb{R}$. In this case, $L = \alpha = \sigma_F = 0$ and $C = L_{G_1}^2 = 2$. We shall illustrate the bound on $\mathrm{KL}(\mu_{\hat{x}^k}|\mu^\star)$ (Cor2 for SPLA and Cor14 of [17] for SSLA) for both algorithms using histograms. Note that the distribution $\mu_{\hat{x}^k}$ of $\hat{x}^k$ is a (deterministic) mixture of the $\mu_{x^j}$: $\mu_{\hat{x}^k} = \frac{1}{k}\sum_{j=1}^k \mu_{x^j}$. Using Pinsker inequality, we can obtain a bound on the total variation distance between $\mu_{\hat{x}^k}$ and $\mu^\star$ from the bound on KL, and this can be illustrated by histograms[1]. In Figure 1, we take $\gamma = 10$ and do $10^5$ iterations of both algorithms. Note that here the complexity of SPLA and SSLA are the same. SPLA enjoys the the well

Figure 1: Comparison between histograms of SPLA and SSLA and the true density $0.5\exp(-|x|)$.

known advantages of proximal methods [41]: precision, numerical stability (less outliers), and robustness to step size.

**R2+R3: Motivations.** There is an abundance of instances of the problem $\min U = \sum_{i=1}^n g_i$, where $n$ is large, each $\mathrm{prox}_{\gamma g_i}$ has a closed form, but $\mathrm{prox}_U$ is intractable (as hard as minimizing $U$); e.g., SVM, logistic regression (see footnote Page 2), overlapping group lasso, TV regularization (see l. 210), see also [16, Section 2]. All these instances can be seen as MAP of $\propto \exp(-U)$ ([21,38,43,46]) and can be tackled by SPLA[1]. For the advantage of sampling a posteriori vs MAP for our example, see [19, Abstract, Paragraph 4.2.1 and 4.2.2]. Sampling allows to avoid overfitting[1].

**R2: Minor comments.** We especially thank R2 for his/her detailed comments. All minor comments will be easily addressed in the camera-ready version of the paper,[1] e.g., we will replace the sketch of the proof by a remark on gradient flows, remarks of R2 on l.382 and Lemma 6 are due to minor typos, and l. 449 and 477 will be easily clarified.

**R3: Trade-offs.** We shall illustrate our answer on l.212. As R3 suggested, $n$ is analogous to the minibatch size in SGD. The larger $n$, the better the approximation of TV by the empirical mean (classical trade-off of SGD). Once $n$ is fixed, one have to choose the level of splitting (*i.e* either treat the full sum in one prox or split each term of the sum). Less splitting is better: splitting is basically approximating the full prox by a combination of many prox (similar to the trade-off of SGD). As R3 says, we don't gain by splitting: one can check that the value of $C$ doesn't change by treating $g$ as $g/2 + g/2$, but in the latter case, two prox are needed at each iteration (so the computation time is twice). However, our key point in this work is that splitting is often unavoidable (see l. 210 and the paragraph "Motivations"). Finally, the value of $C$ is smaller (better) if the noises impacting the $g_i$, $i \geq 2$ are independent[1].

## Footnotes

[1]We will provide more details in the paper/supplementary but not here due to the lack of space.


[Meta-Review · NeurIPS 2019]

This paper should be of significant interest to the community given the novelty of the proposed sampling method, the solid theoretical results, and the applicability to online sampling.